

# Inulin protects against the harmful effects of dietary emulsifiers on mice gut microbiome

Cansu Bekar[1], Ozlem Ozmen[2], Ceren Ozkul[3] and Aylin Ayaz[4]

[1] Department of Nutrition and Dietetics, Burdur Mehmet Akif Ersoy University, Burdur, Turkey
[2] Department of Pathology, Burdur Mehmet Akif Ersoy University, Burdur, Turkey
[3] Department of Pharmaceutical Microbiology, Hacettepe University, Ankara, Turkey
[4] Department of Nutrition and Dietetics, Hacettepe University, Ankara, Turkey

Corresponding author
Cansu Bekar, cansubekar@mehmetakif.edu.tr

## ABSTRACT

**Background**. The prevalence of inflammatory bowel diseases is increasing, especially in developing countries, with adoption of Western-style diet. This study aimed to investigate the effects of two emulsifiers including lecithin and carboxymethyl cellulose (CMC) on the gut microbiota, intestinal inflammation and the potential of inulin as a means to protect against the harmful effects of emulsifiers.

**Methods**. In this study, male C57Bl/6 mice were divided into five groups (n:6/group) (control, CMC, lecithin, CMC+inulin, and lecithin+inulin). Lecithin and CMC were diluted in drinking water (1% w/v) and inulin was administered daily at 5 g/kg for 12 weeks. Histological examination of the ileum and colon, serum IL-10, IL-6, and fecal lipocalin-2 levels were analyzed. 16S rRNA gene V3-V4 region amplicon sequencing was performed on stool samples.

**Results**. In the CMC and lecithin groups, shortening of the villus and a decrease in goblet cells were observed in the ileum and colon, whereas inulin reversed this effect. The lipocalin level, which was $9.7 \pm 3.29$ ng in the CMC group, decreased to $4.1 \pm 2.98$ ng with the administration of inulin. *Bifidobacteria* and *Akkermansia* were lower in the CMC group than the control, while they were higher in the CMC+inulin group. In conclusion, emulsifiers affect intestinal health negatively by disrupting the epithelial integrity and altering the composition of the microbiota. Inulin is protective on their harmful effects. In addition, it was found that CMC was more detrimental to microbiota composition than lecithin.

## INTRODUCTION

Inflammatory bowel diseases are a chronic, heterogeneous group of diseases that cause inflammation in the gastrointestinal tract, including Crohn's disease and ulcerative colitis (*Malik, 2015*). With the increase in the prevalence of inflammatory bowel diseases, they have become an important public health problem worldwide (*Ng et al., 2017*). Although the etiology of inflammatory bowel disease is not completely known, alteration in the gut microbiota composition, called dysbiosis, is thought to be an important factor in pathogenesis (*Khan et al., 2019*). Chronic intestinal inflammation usually occurs in

the terminal ileum and colon, which is the region of the highest intestinal microflora concentration (*Akram, Garud & Joshi, 2019*).

A variety of factors induce microbiota dysbiosis and encroachment, including consumption of the Western-style diet. The Western-style diet includes high amounts of protein, saturated fat, sugar, processed foods, and food additives, while it includes low consumption of vegetables, fruits, and fiber (*Schreiner et al., 2020*). Emulsifiers are a class of food additives used in many processed foods to extend shelf life and improve organoleptic properties. Microbiota dysbiosis caused by emulsifiers is considered to be an important factor in the association between consumption of ultra-processed foods and the development of chronic inflammatory diseases (*Vo, Lynch & Roberts, 2019*).

Recent studies have reported that various food emulsifiers and thickeners such as carboxymethyl cellulose (CMC), polysorbate 80 (P80) and carrageenan increase intestinal permeability, bacterial translocation and the passage of bacterial components such as lipopolysaccharides into the systemic circulation, thus causing intestinal and systemic inflammation (*Shang et al., 2017*; *Viennois et al., 2020*). It has been stated that emulsifiers may play a role in the development of various diseases from inflammatory bowel disease to metabolic syndrome by changing the composition of fecal microbiota, and the amount and composition of short-chain fatty acids (SCFA) (*Chassaing et al., 2015*; *Furuhashi et al., 2020*).

Prebiotics are non-digestible nutritional components usually included in dietary fiber that can be fermented by the gut microbiota and selectively stimulate the growth and/or activity of health-related gut bacteria (*Quigley, 2019*). Inulin is a general term for all linear fructans (inulin–type fructans) with different degrees of polymerization. The length of the fructosyl chains of inulin molecules can range from 2–60 fructose units. The main sources of inulin are chicory, asparagus, onions, garlic, banana, rye, barley, and artichokes. Inulin protects the microbial population, which helps the host to defend against pathogen translocation, supports the epithelial barrier function, and increases the production of SCFA and has positive effects on the prevention of symptoms of IBD and immune modulation (*Akram, Garud & Joshi, 2019*; *Tawfick et al., 2022*).

The protective effect of inulin on the gut microbiome and inflammation caused by emulsifiers have not been investigated before. This study mainly aimed to investigate the effect of CMC and lecithin on intestinal inflammation, as well as the potential of inulin to prevent emulsifier-induced disruption of host microbiota homeostasis and how close inulin get the emulsifier groups to control.

## MATERIALS & METHODS

### Animal experiments

Mice are very suitable to be selected as animal models because of their high gene homology with human genes, and low test cost. In addition, mice are prone to microbiota changes with diet like humans. Five week wild-type C57BL/6 male mice were obtained from Burdur Mehmet Akif Ersoy University Laboratory Animals Production and Experimental Research Center, Turkey and the study was carried out in this center. Mice were

housed under 12-h light/dark cycle at the standard temperature ($22 \pm 2$ °C), and 55–60% relative humidity. All animals were fed with standard chow and water *ad libitum*. All experiments on animals have been carried out taking into account the guidelines for animal research from the ARRIVE (Animal Research: Reporting in Live Experiments) 2.0. This study protocol was approved by Burdur Mehmet Akif Ersoy University Experimental Animals Local Ethics Committee, Turkey (2021, no. 735).

The bedding materials were mixed weekly during a two week acclimation period to minimize the cage effect before being randomly divided into different groups. A total of 30 multiple litters mice were divided equally into five groups (two separate cages per condition) and there were six mice in each group. Mice were randomly separated into control and experimental group. Mice were exposed to water (control group) ($n = 6$), CMC ($n = 6$), lecithin ($n = 6$), CMC+inulin ($n = 6$) or lecithin+inulin ($n = 6$) for 12 weeks. For prebiotic treatment, oligofructose-enriched inulin was given to each mice by oral administration at the dose of 5 g/kg body weight, which was dissolved in water. Oligofructose-enriched inulin is a mixture of long-chain inulin and oligofructose (Beneo Synergy1), was obtained from Orafti1 (Tienen, Belgium). Therefore, no analgesics were used because it was anticipated that these interventions would not cause pain. Neither expected nor unexpected adverse events were observed.

Lecithin and CMC were diluted in drinking water (1% w/v) and the solutions were changed every week. At the end of every week, the average water consumption of the mice in the cage was determined by measuring the remaining amounts for one week. Body weights were measured every week. Food intake was measured every week by placing groups of mice in a clean cage with a known amount of food, 24 h later, the amount of remaining food was measured for daily food intake. The general health status of the mice was checked every day.

At the end of the study, thirty mice were sacrificed by cervical dislocation under general anesthesia which was achieved by intraperitoneally injecting 100 mg/kg ketamine and 5 mg/kg xylazine to each animal. Rapid and severe weight loss, behavioral disorder, and severe diarrhea were the main criteria established for euthanizing animals prior to planned end of the experiment, but this was not needed.

## Sample collection and analysis

After 3 months of emulsifier treatment, fresh feces were collected for analysis of the fecal lipocalin-2 (lcn-2) level and microbiota composition. A blood sample was collected by cardiac puncture under general anesthesia. Each blood sample was centrifuged at 3,500 g for 10 min. The plasma samples were stored at −80 °C until analysed. IL-6 and IL-10, were measured in the serum samples using commercially available ELISA kits according to the manufacturer's instructions. Lcn-2 levels were analysed in the fecal supernatants using the lcn-2 ELISA kit, and the optical density was read at 450 nm.

## Histopathological examination

Small intestine and colon tissue samples were obtained at the end of the study as previously described in *Diler et al. (2021)*. These tissues were fixed in 10% buffered

formalin, and processed using automatic tissue processing equipment (Leica ASP300S; Leica Microsystems, Nussloch, Germany) for histological examination. After paraffin was embedded, the samples were sectioned at 5 μm using a Leica RM 2155 rotary microtome (Leica Microsystems, Nussloch, Germany). Following this, the sections were stained with hematoxylin and eosin (H&E) and examined under a microscope. For morphometric analysis, 10 random villus were selected separately from the ileum and colon of each mouse. The length and thickness of the villus, and the length of the crypt were measured at 40x magnification using an Olympus CX21 light microscope. The Database Manual Cell Sens Life Science Imaging Software System (Olympus Corporation, Tokyo, Japan) was used to carry out the morphometric evaluation. In order to quantify the stained mucous cells, 20X magnification was used to examine three areas from each mouse. Each ileum and colon were assigned scores out of three based on the degree of epithelial damage and inflammatory infiltrate in the mucosa, submucosa, and muscularis/serosa semiquantitatively. According to the severity, the scores were assigned to descriptions ranging from 0 to 3.

Masson trichrome staining was performed on the intestine and colon for evaluation of the amount of connective tissue and collagen by ready-to-use dye kit (Masson Trichrome with aniline Blue, Bio-Optica, Milano, Italy) according to the manufacturer's protocol. The results were evaluated by comparing the blue-stained connective tissue regions between groups.

For PAS staining of goblet cells, ready to use dye kit (Periodic acid Schiff (PAS), Hotchkiss McManus kit for AUS240, Bio-Optica, Milano, Italy) was used according to the manufacturer's protocol. Image J 1.46r (National Institutes of Health, Bethesda MD) program was used to count positive cells in the ileum and colon. For counting, all positive cells in the 20X objective area were clicked on one by one and the program marked and counted these cells. This process was applied to 10 sections for each group in both the ileum and colon.

### DNA isolation, library construction, and 16S rRNA sequencing

DNA was isolated from fecal samples using a DNeasy PowerSoil Pro kit (Qiagen, Hilden, Germany) according to the manufacturers' instructions. The V3-V4 region of the 16S rRNA gene was amplified using primers 341F (5'-CCTAYGGGRBGCASCAG-3') and 806R (5'-GGACTACNNGGGTATCTAAT-3') along with barcode (*Yu et al., 2005*). The presence of unique amplicons was confirmed by detecting the presence of a 466 bp band on 2% agarose gel electrophoresis. DNA libraries were prepared using the Nextera XT DNA Library Preparation Kit (Illumina, San Diego, CA, USA). The library quality was assessed with a Qubit fluorometer (Thermo Scientific, Waltham, MA, USA). Sample libraries were then pooled at an equal molar concentration and sequenced on Illumina NovaSeq 6000 platform, and 250 bp paired-end reads were generated.

### Microbial population analysis

The Quantitative Insights into Microbial Ecology (QIIME-2, version 2022.11) was used to analyze data (*Bolyen et al., 2019*). Sequencing reads were trimmed and denoized using DADA2 (*Callahan et al., 2016*). A phylogenetic tree was generated using MAFFT alignment

(*Katoh & Standley, 2013*). The taxonomy was assigned using a naïve Bayes classifier trained on the Greengenes 13_8 99% full-length sequences database. ASVs that were present in ≥10 samples and had a total count of >15 across all samples were included in subsequent analysis. *Cyanobacteria*, mitochondrial, and chloroplast sequences were filtered from the feature tables and sequences. Alpha diversity (observed_features, Pielou's evenness, Shannon, faith_pd) was determined for microbial community analysis.

## Statistical analysis

All statistical analyses were performed using Statistical Package for the Social Sciences (SPSS) software (IBM SPSS Statistics for Windows, Version 23.0. Armonk, NY, USA). The conformity of the data to the normal distribution was evaluated with Kolmogorov–Smirnov/Shapiro–Wilk test and histogram graphs. Statistical analyses were performed by one-way analysis of variance (ANOVA) to compare the means of normally distributed parameters between groups, and Kruskal-Wallis tests were used for not normally distributed parameters. All values are expressed as the mean ± standard deviation. *P*-value of less than 0.05 was considered to show a statistically significant result. Bray-Curtis and unweighted UniFrac measurements were calculated for $\beta$-diversity and visualized with Principal Coordinate Analysis (PCoA). Permutational multivariate analysis of variance (PERMANOVA) was used to compare distances. The linear discriminant analysis effect size (LEfSe) was used to assess the significantly differentiated taxa between groups (*Segata et al., 2011*). A Linear discriminant analysis (LDA) cutoff >2.0 with a *P*-value <0.05 was considered significant.

## RESULTS

### Food consumption and weight gain

Table 1 showed the body weight, consumption of food, water and emulsifiers of mice. There was no significant difference between the groups in terms of initial weight, weight gain, water and emulsifiers consumption of mice (*P*>0.05). Food consumption of the mice was lower in the CMC+inulin (2.9 ± 0.66 g/day) and lecithin+inulin groups (3.1 ± 0.63 g/day) than in the CMC, lecithin and control groups (3.5 ± 0.59 g/day, 3.4 ± 0.45 g/day, 3.9 ± 0.64 g/day, respectively) (*P* = 0.001).

### Intestinal histopathology and inflammation

Histopathological examination of the small intestine revealed a normal microscopic appearance in the control group. In the CMC and lecithin groups, shortening and bulking of the villus were observed in the ileum of the mice. In the CMC+inulin and lecithin+inulin groups, elongation was observed in the villus. Intestinal flora was significantly available in the intestinal lumen in the CMC and lecithin groups, but this microbial migration did not increase serum inflammatory parameters. Inulin administration decreased amount of bacteria in both the CMC and lecithin groups, but did not result in complete improvement (Fig. 1). The results of statistical analysis of histopathological score, villus length, crypt length, and villus thickness are shown in Table 2.

Masson's trichrome staining of the ileum and colon revealed a decrease in the amount of collagen in the connective tissue in the CMC and lecithin groups compared to the

**Table 1** Body weights, food, water and emulsifiers consumption of the groups (mean ±SD).

| Parameters | Groups (n:6) | | | | | |
| --- | --- | --- | --- | --- | --- | --- |
| | Control | CMC | Lecithin | CMC+Inulin | Lecithin+Inulin | *P* value |
| Body weights (baseline) (g) | 16.8 ± 2.16 | 16.1 ± 2.01 | 17.2 ± 2.87 | 17.3 ± 2.11 | 16.0 ± 2.40 | 0.808 |
| Changes in body weight (%) | 70.7 ± 24.28 | 66.4 ± 19.10 | 54.2 ± 13.97 | 55.3 ± 20.20 | 51.9 ± 8.75 | 0.308 |
| Food (g/day) | 4.0 ± 0.59[a] | 3.6 ± 0.59[a] | 3.5 ± 0.30[a] | 2.9 ± 0.66[b] | 3.1 ± 0.64[b] | <0.001[**] |
| Water (ml/day) | 7.3 ± 1.39 | 6.7 ± 0.84 | 7.6 ± 1.08 | 7.0 ± 1.17 | 7.8 ± 1.11 | 0.199 |
| Emulsifiers (mg/day) | – | 67.3 ± 8.43 | 75.6 ± 10.81 | 70.3 ± 11.68 | 77.6 ± 11.12 | 0.083 |

**Notes.**

[a, b]There are statistical differences between the means having the different superscripts on the same line.

[**]*p* < 0.001.

CMC, Carboxymethyl cellulose.

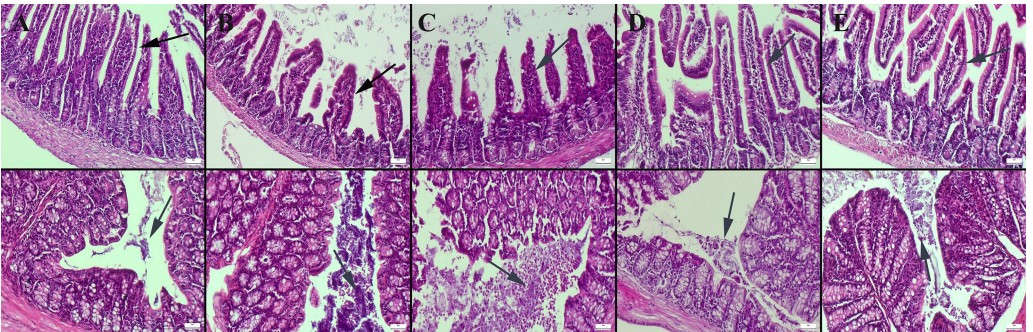

**Figure 1** Histopathological examination findings of ileums (upper row) and colons (below row) between groups. (A) Normal intestinal histology in ileum and normal intestinal flora in colon in the control group, (B) ileum with enlarged lumen and shortened villus and markedly increased amount of bacteria in the colon lumen in the CMC group, (C) shortened and fused villus in the ileum and significantly increased amount of bacteria in colon lumen in the lecithin group, (D) elongated villus in ileum and significant decrease in amount of bacteria in colon in the CMC+inulin group, (E) elongated villus (arrows) and significant decrease in amount of bacteria in colon lumen in the lecithin+inulin group (arrows), HE, bars = 50 μm. CMC: Carboxymethyl cellulose.

control group, and it was observed that the administration of inulin slightly normalized the connective tissue (Fig. 2). In PAS staining, the number of goblet cells decreased significantly in the CMC and lecithin groups in the ileum and colon. Inulin administration increased the number and size of goblet cells in both CMC+inulin and lecithin+inulin groups (Fig. 3). There was no significant difference between the groups in terms of serum cytokines IL-6 and IL-10 and fecal levels of the inflammatory marker (*P*>0.05) (Table 2). However, lipocalin levels, which were higher in CMC group (9.7 ± 3.29 ng) compared to control group (4.7 ± 3.49 ng) decreased to 4.1 ± 2.98 ng when inulin was administered, and the result was significant in the pairwise comparison (*P* = 0.01).

## Fecal microbiota analysis

A total of 1,148,814 high-quality reads were obtained from 25 samples (ranging from 28,345 to 60,283 reads per sample). No significant difference was observed for ASVs and Shannon index, however, both tended to be lower, especially in CMC group. Faith's
**Table 2   Histopathological examination and inflammatory markers of groups (mean ± SD).**

| | Sample | Groups (n:6) | | | | | |
| --- | --- | --- | --- | --- | --- | --- | --- |
| | | Control | CMC | Lecithin | CMC+Inulin | Lecithin+Inulin | *P* value |
| Histopathology scores | Ileum | $0.0 \pm 0.0^a$ | $1.0 \pm 0.25^b$ | $1.0 \pm 0.25^b$ | $0.5 \pm 0.22^a$ | $0.7 \pm 0.21^b$ | $<0.01^{**}$ |
| | Colon | $0.2 \pm 0.16^a$ | $1.3 \pm 0.21^b$ | $1.7 \pm 0.21^b$ | $0.5 \pm 0.22^{ac}$ | $0.8 \pm 0.16^c$ | $<0.001^{***}$ |
| Villus length (µm) | Ileum | $399.8 \pm 9.58^a$ | $265.7 \pm 10.36^b$ | $250.3 \pm 7.64^b$ | $291.0 \pm 6.25^c$ | $298.8 \pm 4.79^c$ | $<0.001^{***}$ |
| | Colon | $103.0 \pm 4.65^a$ | $83.3 \pm 1.11^b$ | $77.8 \pm 3.12^b$ | $90.3 \pm 0.91^c$ | $87.0 \pm 6.16^c$ | $<0.001^{***}$ |
| Villus thickness (µm) | Ileum | $43.5 \pm 2.20^a$ | $56.2 \pm 1.07^b$ | $57.5 \pm 1.17^b$ | $43.3 \pm 0.66^a$ | $41.7 \pm 1.25^a$ | $<0.001^{***}$ |
| | Colon | $48.7 \pm 1.72^a$ | $54.3 \pm 0.55^c$ | $53.3 \pm 1.74^{bc}$ | $49.2 \pm 0.70^a$ | $50.0 \pm 0.70^{ab}$ | $<0.001^{***}$ |
| Crypt lenght (µm) | Ileum | $67.3 \pm 2.94^a$ | $42.8 \pm 3.06^b$ | $30.7 \pm 2.58^c$ | $52.7 \pm 2.80^d$ | $57.8 \pm 1.47^e$ | $<0.001^{***}$ |
| | Colon | $27.8 \pm 5.49^a$ | $15.3 \pm 2.50^{bc}$ | $11.7 \pm 1.86^b$ | $17.0 \pm 2.09^c$ | $16.8 \pm 1.94^c$ | $<0.001^{***}$ |
| Goblet cell count | Ileum | $55.8 \pm 2.24^a$ | $40.5 \pm 0.88^b$ | $42.3 \pm 1.14^b$ | $54.5 \pm 0.56^a$ | $51.0 \pm 2.38^a$ | $<0.001^{***}$ |
| | Colon | $108.8 \pm 2.35^a$ | $89.0 \pm 0.63^b$ | $92.5 \pm 1.47^b$ | $102.5 \pm 1.23^c$ | $103.5 \pm 1.66^c$ | $<0.001^{***}$ |
| Lcn-2 | Stool | $4.7 \pm 3.49$ | $9.7 \pm 3.29$ | $5.7 \pm 3.77$ | $4.1 \pm 2.98$ | $6.2 \pm 3.54$ | $0.077$ |
| IL-6 | Serum | $105.8 \pm 8.41$ | $96.3 \pm 11.73$ | $101.5 \pm 10.10$ | $111.6 \pm 2.98$ | $102.3 \pm 7.42$ | $0.063$ |
| IL-10 | Serum | $261.2 \pm 40.52^a$ | $179.5 \pm 29.47^b$ | $186.1 \pm 25.1^b$ | $238.6 \pm 40.52^a$ | $220.8 \pm 32.58^a$ | $0.001^{**}$ |

**Notes.**

[a, b, c, d]There are statistical differences between the means having the different superscripts on the same line.

[**]$p < 0.05$.

[***]$p < 0.001$.

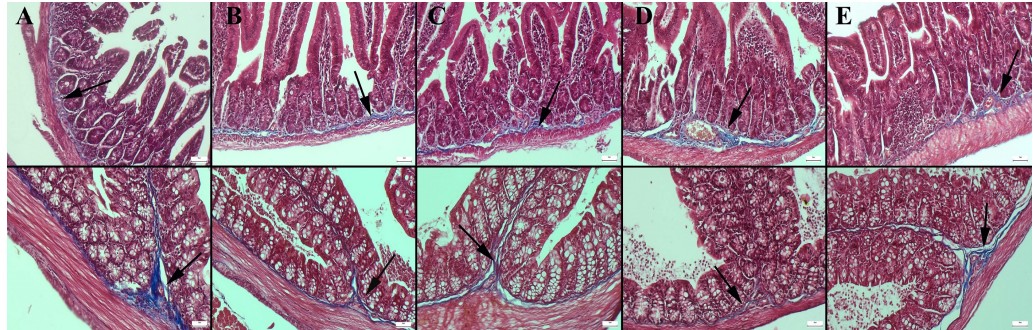

**Figure 2   Masson trichrome staining in the ileum (upper row) and colon (below row).** (A) The prominent connective tissue and collagen in the control group, (B) the thinning of the connective tissue in the submucosa in the CMC group, (C) the thinning of the connective tissue and collagen in the lecithin group, (D) connective tissue layer thickening in the CMC+inulin group, (E) slight increase in the connective tissue layer (arrows) in the lecithin+inulin group, Masson trichrome staining, bars =50 µm, CMC: Carboxymethyl cellulose.

phylogenetic diversity was significantly lower in CMC group compared to the lecithin and CMC+inulin groups ($P = 0.028$ for both comparisons; Fig. 4A). This result indicates that CMC administration decreased the phylogenetic diversity, while inulin administration leads to a significant increase in Faith's pd compared to the CMC group.

Microbial community structure were analysed by employing Bray-Curtis dissimilarity which is based on abundance and unweighted UniFrac distance which encounters the presence/absence of ASVs. Results showed that all groups were clustered separately, which

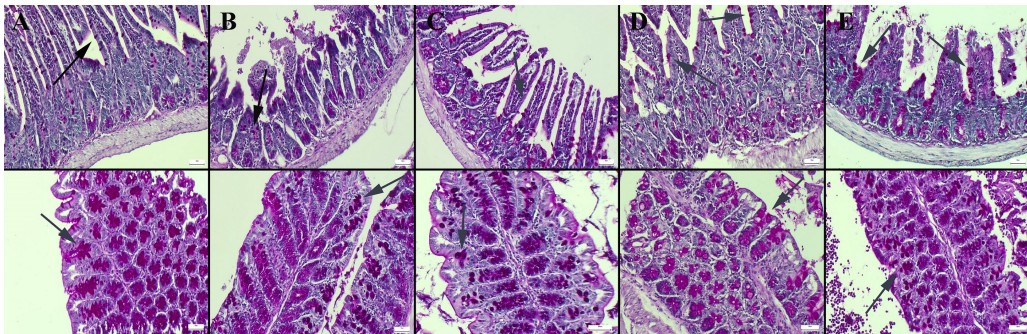

**Figure 3  Periodic Acid Schiff staining of ileum (upper row) and colon (below row).** (A) Normal appearance and number of goblet cells in the control group, goblet cells (B) significantly reduced in the CMC group, (C) decreased in the lecithin group, (D) increased and enlarged in the CMC+inulin group, (E) markedly increased and enlarged in the lecithin+inulin group (arrows), PAS staining, bars =50 μm. CMC: Carboxymethyl cellulose.

was more pronounced in both inulin-administered groups according to Bray-Curtis dissimilarity (PERMANOVA, $P < 0.05$; Fig. 4B). Unweighted UniFrac distances also revealed significant differences for all comparisons except for CMC+inulin *vs.* control group, which may be explained by a shift from a CMC-associated microbiome to control group after inulin administration (PERMANOVA, $P < 0.05$; Fig. S1A, Table S1).

It was observed that *Bacteroidetes* phylum was dominant in all groups, followed by *Firmicutes*, and the rest consisted of *Campylobacterota, Actinobacteria, Verrucomicrobia, Proteobacteria* and *Patescibacteria* phylum. *Actinobacteria* and *Verrucomicrobia* phylum were found to be higher in CMC+inulin group than in the other groups (Fig. 4C). *Muribaculaceae, Prevotellaceae, Lachnospiraceae, Helicobacteraceae,* and *Bacteroidaceae* were the most abundant families among all groups. CMC exposure and inulin administration led to an increase in *Prevotellaceae* levels compared to the control and lecithin groups (Fig. S2).

We used the LEfSe algorithm to determine significantly differentiated taxa between groups. In general, the gut microbiome in the lecithin group was likely less affected compared to CMC group (Figs. 5A–5B, Figs. 6A–6B). *Marvinbryantia,* a member of *Lachnospiraceae,* and *Fournierella*, a member of *Ruminococcaceae* family significantly increased and *Lactobacillus* significantly decreased in the CMC group compared to the control group (Fig. 5A, Table S2). Lecithin group showed a significant enrichment in *Eubacterium ruminantium* and decrease in *Bifidobacterium* compared to the control group (Fig. 5B). In addition, the increased *Helicobacter* in the lecithin group decreased with the administration of inulin (Fig. 6B, Fig. S1B). CMC reduced *Actinobacteria, Bifidobacterium, Lactobacillus, Verrucomicrobiota*, and *Akkermansia*, whereas administration of inulin reversed this effect (Fig. 4C, Fig. 6A, Table S2). Administration of inulin resulted in an increase in *Prevotellaceae NK3B31, Prevotellaceae UCG 001, Gordonibacter, Christensenellaceae, Clostridia UCG 014, Eubacterium nodatum group,* and *Peptostreptococcales* in both CMC and lecithin groups (Figs. 6A–6B, Table S2).

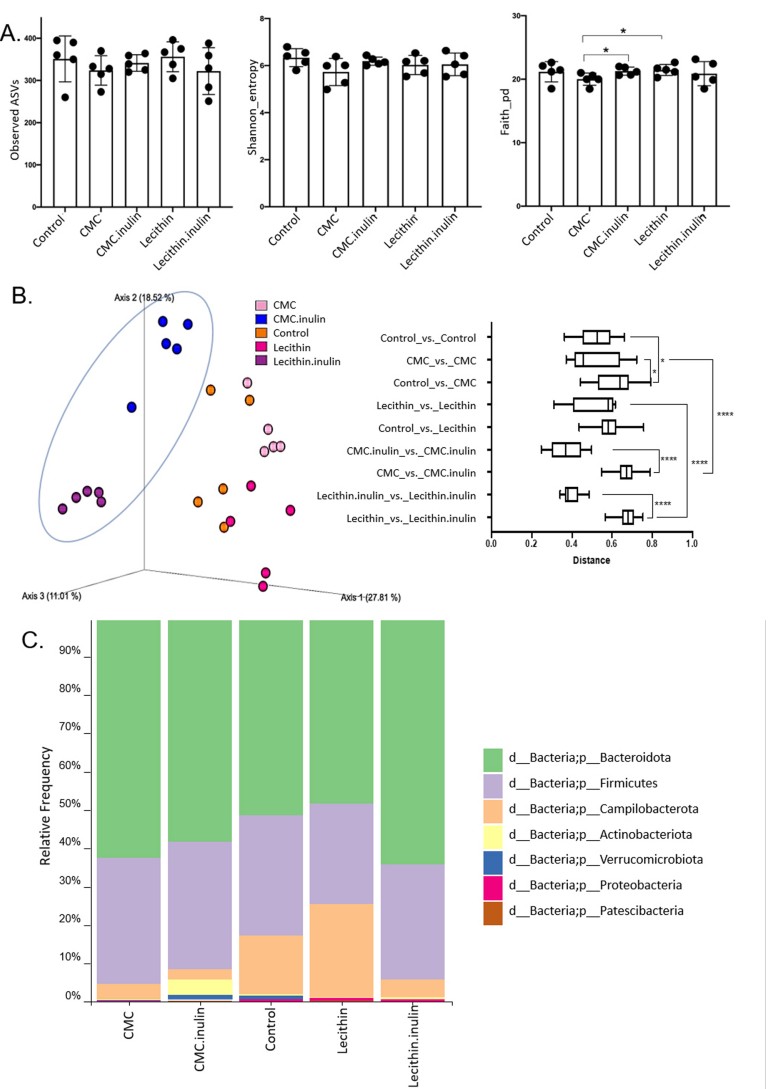

**Figure 4** **Microbial community analysis between the groups.** (A) Alpha diversity metrics including Observed ASVs, Shannon, Faith_pd between the groups (B) Principal coordinates analysis (PCoA) of ASVs based on Bray-Curtis dissimilarity between groups and inter- and intra- group distances based on Bray–Curtis dissimilarity (C) Taxa abundances in phylum level. Group comparisons were performed by ANOVA and Kruskal-Wallis $*p < 0.05, ****p < 0.0001$.

## DISCUSSION

The Western-style diet is characterized by a high intake of fat, sugar, protein, processed foods, and low dietary fiber. Processed foods are industrial formulations that contain many food additives, including flavor enhancers, colorants, and emulsifiers (*Monteiro et al., 2018*). It has been reported that emulsifiers and thickeners may have health effects by disrupting the barrier function of intestinal mucosa, alterations the functional properties of the gut microbiota, and causing mucosal inflammation (*Halmos, Mack & Gibson, 2019*). Prebiotics play an important role in regulating the microbiota by inhibiting the

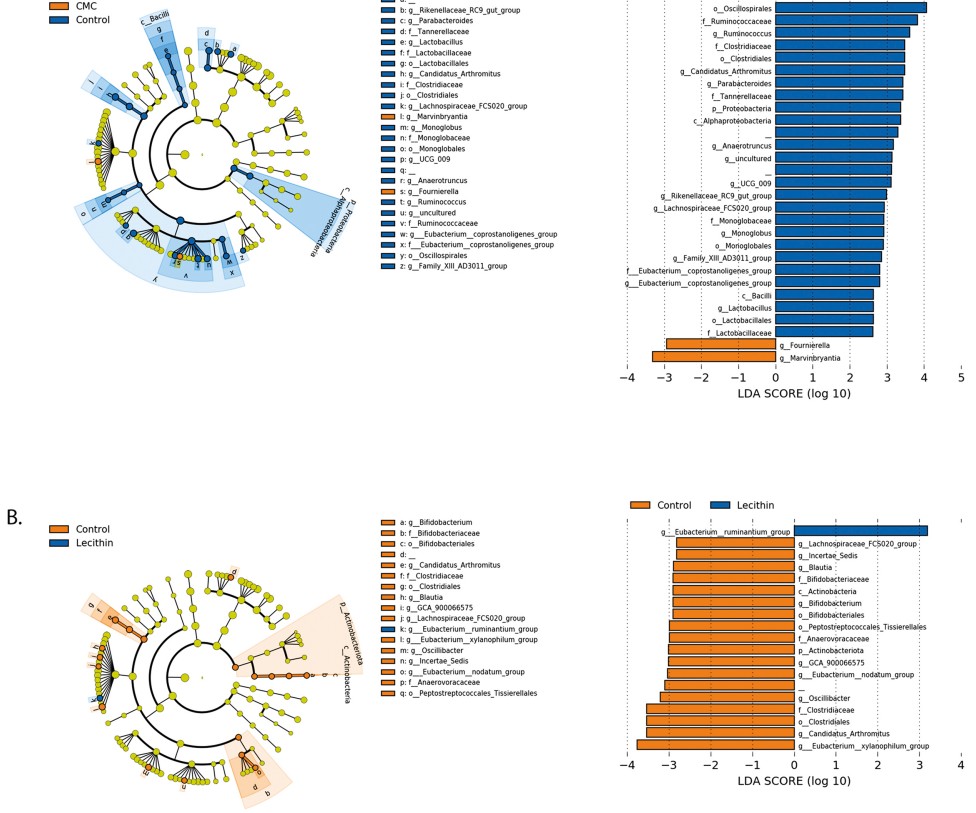

**Figure 5** **LEfSe analysis indicating the significantly differentiated taxa for CMC and lecithin group.** (A) Between control and CMC groups, (B) between control and Lecithin groups. A LDA cutoff > 2.0 with a *p* value < 0.05 was considered significant.

production of SCFA, the proliferation of beneficial bacteria, and preventing colonization of pathogenic microorganisms (*Sanders et al., 2019*). This study examined the effects of different emulsifiers on the intestinal microbiota and intestinal inflammation and the potential of inulin as a means of preventing disruption of host microbiota homeostasis. Our primary hyphothesis was to test whether inulin administration improve emulsifier-induced microbial community structure by getting it closer to that of control mice.

High dietary fiber intake has been associated with lower body weight and prevention of weight gain. These effects have been reported to be due to supresion of hunger, reduction in food intake, alterations in microbiota, acceleration of gastrointestinal transit, reduction in nutrient absorption, and changes in SCFA (*Chen et al., 2018*; *Schäfer et al., 2021*). It has been reported that inulin administration reduced food intake and weight gain in mice with high-fat diet and a streptozotocin-induced diabetes model (*Shao et al., 2020*). In our study, the inulin groups receiving both CMC and lecithin had lower weight gain than the ones in CMC- and lecithin-only groups, in parallel with the decrease in food intake. Consistent with a previous study in which mice were administered CMC, P80, lecithin, and gum arabic, no significant difference in food intake was observed between CMC, lecithin,

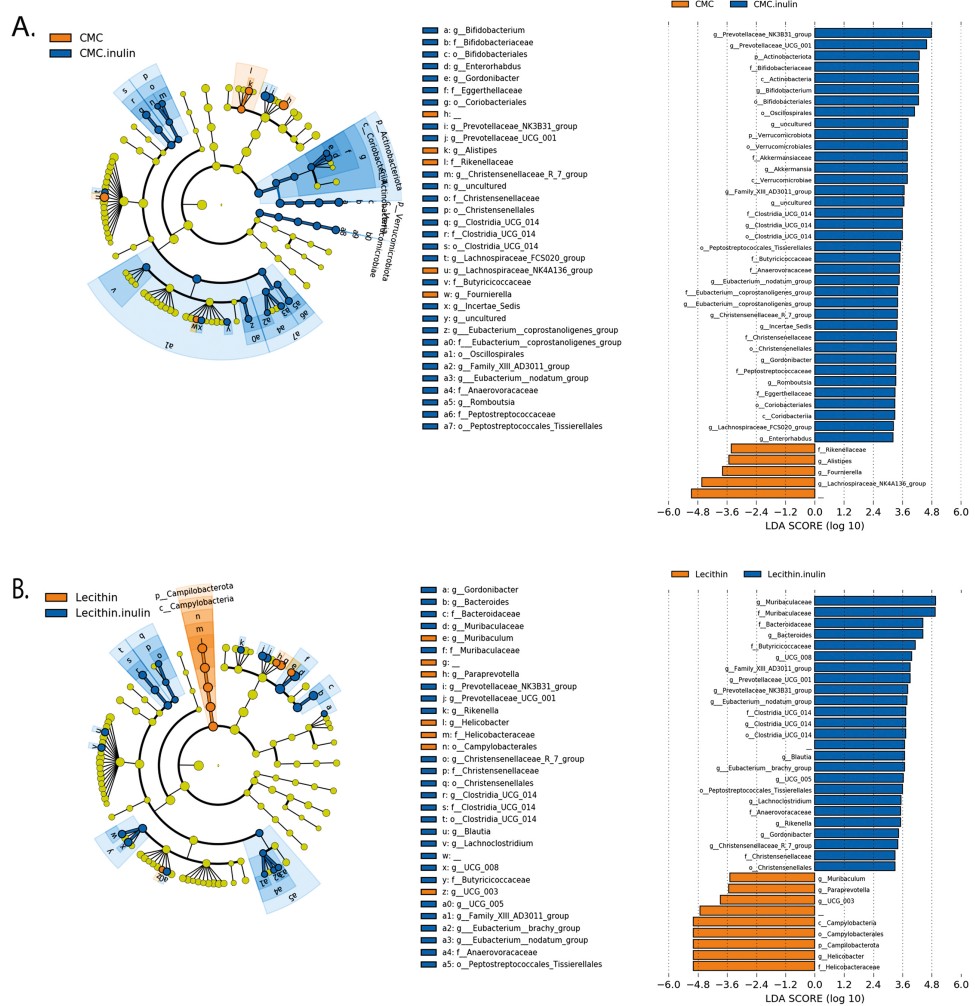

**Figure 6** **LEfSe analysis indicating the significantly differentiated taxa for CMC and lecithin added inulin group.** (A) After inulin administration to CMC groups (B) after inulin administration to Lecithin groups. A LDA cutoff > 2.0 with a *p* value < 0.05 was considered significant.

and control groups (*Sandall et al., 2020*). We found that the difference in weight gain between the groups was not statistically significant as in previous studies (*Daniel, Gewirtz & Chassaing, 2023*).

The disrupted mucosal barrier promotes intestinal inflammation by increasing intestinal permeability and exposure to luminal contents, and initiating an immunological response (*Michielan & D'Incà, 2015*). When mice genetically predisposed to inflammatory bowel disease (IL-10 gene-deficient) were administered a 2% w/v CMC solution for 3 weeks, increased bacterial growth in the intestinal mucosa, decreased villus length, increased inter-villus space, migration of bacteria to the bottom of the crypts, and increased bacterial adhesion to the mucosa, which were similar to inflammatory bowel disease in humans were observed (*Swidsinski et al., 2009*). Similarly, in another study it was reported that CMC and P80 induced colitis in genetically susceptible mice, but not in wild-type. In

addition, they led to chronic intestinal inflammation, shortened the colon and damage to the epithelium in both types mice. It has been reported that emulsifier exposure increased bacterial adhesion in the colon twofold, reduced the contact distance of some bacteria to the epithelium, and this invasion of the microbiota was associated with a decrease in mucus thickness (*Chassaing et al., 2015*).

In agreement with previous studies, (*Chassaing et al., 2015*; *Swidsinski et al., 2009*) it was found that both CMC and lecithin caused shortening and bulking of the villus in the ileum. Intestinal flora was significantly available in the intestinal lumen in the CMC and lecithin groups, but this microbial migration did not increase serum inflammatory parameters. In a similar study, CMC was reported to increase the immigration of microbiota into the epithelium, which was not associated with serum inflammatory parameters (*Daniel, Gewirtz & Chassaing, 2023*). We evaluated the effect of inulin on this condition for the first time, and observed that administration of inulin elongated the villus and decreased amount of bacteria in the colon lumen in both the CMC and lecithin groups, but did not result in complete improvement.

Mucus is a viscous fluid rich in mucin glycoprotein secreted by goblet cells. Mucus secretion from goblet cells is regulated by intestinal microorganisms and host-spesific metabolites. Mucin-2 produced by goblet cells organizes the mucus layer and transmits tolerogenic signals to dendritic cells, limiting the immunogenicity of intestinal antigens (*Okumura & Takeda, 2018*). CMC and P80 have been reported to damage to the mucus layer, increase bacterial adhesion, and have a more proinflammatory microbiota composition (*Chassaing et al., 2015*). In a recent study, the composition of the microbiota altered in terms of mucin-destroying bacteria, the colon mucus became thinner, and intestinal permeability increased in mice given CMC for 12 weeks. When mice were administered galactooligosaccharide to improve the damaged intestinal microbiota, mucin-2 gene expression increased, mucus layer thickness and intestinal barrier function were improved (*Xu et al., 2019*). In another study, the number of goblet cells decreased in mice given CMC and P80, and the mucus barrier was strengthened when *A. muciniphila* was given (*Daniel, Gewirtz & Chassaing, 2023*). Also, in this study, it was found that the number of goblet cells decreased in the CMC and lecithin groups when compared with the ones in control group. As a new solution suggest, we administered the inulin to CMC and lecithin groups and observed increases the number and size of goblet cells. Similarly, it was reported that cecal mucin levels increased in mice given 2.5g/kg and 5g/kg body weight of standard inulin or short chain inulin by increasing the relative abundance of butyrate-producing bacteria (*Zhu et al., 2017*).

An important consequence of inflammatory bowel disease is intestinal fibrosis. As fibrosis progresses, some segments of the intestine become narrowed, which affect severely the structure and functionality of the intestine and worsen the patients' quality of life. An essential fibrosis-initiating factor is chronic inflammation. Inflammatory bowel disease frequently results in intestinal fibrosis, which is typically described as excessive buildup of scar tissue in the gut wall (*Wang et al., 2022*). In our study, no fibrosis was observed, even a slight decrease in the amount of collagen in the connective tissue of the propria mucosa

was observed in the CMC and lecithin groups compared with the control group. This may indicate that intestinal permeability increases in these groups.

It was reported that fecal lipocalin-2 levels in mice given CMC and P80, increased only slightly in wild-type mice, but increased sharply (approximately 10-fold) in genetically susceptible mice. Emulsifiers did not cause significant colitis in nongenetically susceptible mice, but caused epithelial damage and changes in colon morphology and low-grade inflammation (*Chassaing et al., 2015*). In another study, dietary CMC intake increased lipocalin levels compared to the control (9.5 ± 2.8 *vs* 2.7 ± 0.8 ng/g feces, respectively). In addition, it was found that inflammation decreased when galactooligosaccharides were administered to mice to improve the damaged gut microbiota (*Xu et al., 2019*). In this study, there was no statistically significant difference between the groups in terms of lcn-2, but the increased lcn-2 in the CMC group compared to the control (9.7 ± 3.29 *vs* 4.7 ±3.4 9 ng, respectively) was decreased with inulin administration (4.1 ± 2.98 ng) ($P = 0.012$). Similarly, it was reported that addition of *A. muciniphila* decreased lcn-2 levels in mice given CMC (*Daniel, Gewirtz & Chassaing, 2023*). In human study, it was observed that administration of 15g/day CMC does not affect levels of lipopolysaccharide and flagellin in feces and does not cause a difference in serum inflammatory parameters (*Chassaing et al., 2022*). In this study, no significant difference was found between groups in serum levels of IL-6 and IL-10 in mice.

Intestinal dysbiosis can cause impaired mucosal immune response and production of proinflammatory cytokines in genetically susceptible hosts, contributing to inflammation in inflammatory bowel diseases (*Amoroso et al., 2020*). The effects of twenty different emulsifiers on the microbiota were invastigated in an ex vivo study, and reported that CMC and P80 had a significant effect on the microbiota, whereas lecithin had a mild effect, and there was no significant difference between the groups in terms of alpha diversity (*Naimi et al., 2021*). In this study, it was observed that there was no significant difference between the groups in terms of alpha diversity, but the CMC changed the community structure more. Microbial community composition changed significantly after inulin administration in both the lecithin and CMC groups, but was more obvious in the CMC group. Moreover, the inulin effect was also higher in the CMC group in terms of the richness and evenness of the microbial community. Indeed, this might be due to the CMC-induced major alterations in the gut microbiome. From an ecological perspective, lower colonization resistance in CMC-associated microbiome could lead to this remarkable change in gut microbiome after inulin administration, as the major taxonomic groups such as *Clostridiales* and *Lactobacillales* decreased upon CMC exposure (*Caballero-Flores, Pickard & Núñez, 2022*). Qualitatively unweighted UniFrac distance measurements showed that community structure changed significantly in both CMC and lecithin groups compared to the control. A similar study reported that CMC and P80 exposure differed from the ones in control group in terms of unweighted UniFrac distance measurements (*Daniel, Gewirtz & Chassaing, 2023*).

In the gut microbiota, *Firmicutes* and *Bacteroidetes* are the most abundant phyla, whereas the phyla *Actinobacteria*, *Proteobacteria,* and *Verrucomicrobia* consisted of the rest (*Illian, Brambilla & Parolini, 2020*). Similar results were also found in this study.

In a study examining the effects of various emulsifiers on the microbiota composition taxonomically, CMC and lecithin were reported to cause a decrease in the presence of *Verrucomicrobia* and *Akkermensia* and an increase in the *Lachnospiraceae* family (*Naimi et al., 2021*). In a human study, consumption of 15g CMC in healthy adults increased postprandial abdominal discomfort and altered the composition of the gut microbiota, decreased *Faecalibacterium prausnitzi* and *Ruminococcus spp.,* and increased the presence of *Lachnospiraceae.* The study also showed that the response of the microbiota to this food additive can be highly individualized (*Chassaing et al., 2022*). Oligofructose-enriched inulin (15 g/day) decreased disease severity, and increased fecal butyrate levels, fecal abundance of *Bifidobacteriaceae*, *Ruminococcaceae*, and *Faecalibacterium* in patients with mild and moderately active ulcerative colitis (*Valcheva et al., 2019*). It was reported that in mice fed a high-fat diet, *Lachnospiraceae* increased, *Bifidobacterium* and *Lactobacillus* decreased, and the administration of both short-chain and long-chain inulin reversed this effect. In addition, the presence of *Helicobacter* was found to increase with consumption of high-fat diet and to be positively correlated with endotoxin, and *Helicobacter* decreased with administration of inulin (*Li et al., 2020*). *Actinobacteria*, including *Bifidobacteria,* is known to have a beneficial effect on the host health. A decrease in *Bifidobacteria* has been associated with inflammatory bowel disease (*Binda et al., 2018*). In our study, CMC reduced *Actinobacteria*, *Bifidobacterium, Lactobacillus, Verrucomicrobiota*, and *Akkermansia*, whereas administration of inulin reversed this effect. *Marvinbryantia,* which is known to be a cellulolytic member of *Lachnospiraceae* (*Wolin et al., 2003*) and *Fournierella* belongs to *Ruminococcaceae* family which were also known for their ability to ferment cellulose (*Flint et al., 2012*) were significantly increased in the CMC group compared to control. In addition, it was observed that the increase in *Helicobacter* in the lecithin group decreased with the administration of inulin. A recent study reported that in mice given CMC and P80, the level of *A. muciniphila* decreased compared to the control, resulting in inflammation in the colon, especially infiltration of inflammatory cells in the mucosa and submucosa. Administration of *A. muciniphila* reduced low-grade intestinal inflammation caused by emulsifiers and altered the microbiota composition (*Daniel, Gewirtz & Chassaing, 2023*). *A. muciniphila* is abundant in the intestinal tract and uses mucin as a source of carbon, nitrogen, and energy. *A. muciniphila*, the only genus belonging to the *Verrucomicrobia* phylum, increases the thickness of the intestinal mucosal layer and improves intestinal barrier function (*Zhai et al., 2019*).

It has been reported that SCFAs are significantly positively correlated with *Prevotellaceae UCG 001*, *Parabacteroides,* and *Bacteroides* (*Zhu et al., 2022*). *Prevotella* genus is more abundant in individuals who eat a diet rich in plant sources such as a Mediterranean and a vegetarian diet (*Ley, 2016*). In our study, *Prevotellaceae NK3B31*, *Prevotellaceae UCG 001*, *Clostridia UCG 014*, and *Peptostreptococcales* were found to change significantly after inulin administration in both lecithin and CMC groups. In addition, *Prevotellaceae* was higher in the CMC+inulin and CMC groups compared to lecithin groups. Another study reported that prophylactic administration of turmeric polysaccharide ameliorated the dextran sulfate sodium-induced gut microbiota imbalance in favor of *Clostridia UCG 014,* which produces the tryptophan metabolite indoleacrylic acid (*Yang et al., 2021*). *Peptostreptococcus* species

also produce indoleacrylic acid, which promotes barrier function of intestinal epithelium and attenuates inflammatory responses. In IBD patients, the ability of microbes to degrade mucins and metabolize tryptophan was decreased (*Wlodarska et al., 2017*).

We acknowledge the following limitation. We evaluated the effects of only two emulsifiers in this study. Generalization of the microbiome-specific effect can be concluded by evaluating the effect of other emulsifiers together with dose-dependent experiments.

## CONCLUSION

In conclusion, CMC and lecithin, which are commonly used as emulsifiers in processed foods, disrupt the intestinal barrier function and cause intestinal inflammation by alterations in the composition of the fecal microbiota. Carboxymethyl cellulose was more effective than lecithin on microbiota composition. The composition of the microbiota, particularly deteriorated by CMC exposure, was improved by administration of inulin. Although the effects of different emulsifiers have been shown in the current literature, studies on lecithin are quite limited. Additionally, nutritional components are needed to regulate the disrupted microbiota and reverse these effects. This is the first time, the benefical effect of inulin on microbiota dysbiosis caused by emulsifiers was investigated. Further clinical studies are needed to confirm these findings to the prevention of the development and progression of inflammatory bowel diseases in human.

### Funding
This research was supported by the Scientific and Technological Research Council of Turkey (TUBITAK) (No. 221S218). The funders had no role in study design, data collection and analysis, decision to publish, or preparation of the manuscript.

### Grant Disclosures
The following grant information was disclosed by the authors:
The Scientific and Technological Research Council of Turkey (TUBITAK): No. 221S218.

### Competing Interests
The authors declare there are no competing interests.

### Author Contributions
- Cansu Bekar conceived and designed the experiments, performed the experiments, analyzed the data, prepared figures and/or tables, authored or reviewed drafts of the article, and approved the final draft.
- Ozlem Ozmen performed the experiments, analyzed the data, prepared figures and/or tables, authored or reviewed drafts of the article, and approved the final draft.
- Ceren Ozkul performed the experiments, analyzed the data, prepared figures and/or tables, authored or reviewed drafts of the article, and approved the final draft.
- Aylin Ayaz conceived and designed the experiments, analyzed the data, prepared figures and/or tables, authored or reviewed drafts of the article, and approved the final draft.

## Animal Ethics

The following information was supplied relating to ethical approvals (*i.e.*, approving body and any reference numbers):

This study was approved by Burdur Mehmet Akif Ersoy University Experimental Animals Local Ethics Committee, Turkey (2021, no. 735)

## Data Availability

The 16S rRNA sequencing data are available at SRA: PRJNA1036446.

## Supplemental Information

Supplemental information for this article can be found online at http://dx.doi.org/10.7717/peerj.17110#supplemental-information.

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
