# Peer review of "Inulin protects against the harmful effects of dietary emulsifiers on mice gut microbiome"

_PeerJ, doi:10.7717/peerj.17110_

## Round 0.1 · original submission · Major Revisions

Upon receiving two reviews that detail serious consideration of the current submission and suggest major revision before consideration for publication, I am inclined to agree that the current submission needs;

1. Solid Bd valid rationale for the choice of biomarkers.

2. Detailed revision of the stats and analyses and subsequent figures used in the study.

3. Detailed justification for not doing a pre and post microbiota analyses, and if just doing one post study is suitable.

Please address the concerns of the reviewers as much as possible where you can and provide a suitable and detailed reason for not being able to where it is necessary.

Kindly resubmit for consideration upon revision.
Sincerely

Reviewer 1 ·

Basic reporting

This manuscript examined the notion that the pre-biotoc fiber, inulin, might protect against detrimental consequences of dietary emulsifiers, CMC and lecithin. The idea is reasonable and the data obtained is consistent with this notion but the study design has significant problems that make the overall value of the observations rather minimal.

Experimental design

1. Lack of an inulin only group makes interpretation much harder. Can’t tell if inulin impacted measured parameters independent of emulsifiers or prevented impacts of these compounds.
2. Very unclear how inulin was administered on a gm/kg basis as stated given that it was added to drinking water, a mode of administration that for inulin differs from other studies further necessitating the inclusion of an inulin group.
3. Discussion of “microbiota density” from formal-fixed histology is pretty hard to make sense of. First of all, the term microbiota density is accepted to refer to # of bacteria per unit mass (typically feces or tissue). Inulin, being prebiotic, is accepted to increase bacterial density. I think the authors are trying to assess bacterial localization but this needs to be done using carnoy fixation to be of value as formalin destroys mucus structures.
4. Re microbiota analysis, they really should have assayed this from feces collected before and after the intervention for all groups. Else, some of the observed differences might have been pre-existing cage to cage variation. I don’t doubt the inulin differences but the others could well be caging differences. Again, what would really matter if CMC or lecithin no longer shifted microbiota in presence of inulin; i.e. CMC/inulin was similar to inulin only but again the latter group was not included making this impossible.

Validity of the findings

hard to conclude anything with confidence given the poor design although I would have a weak guess that the central hypothesis might have been correct.

Reviewer 2 ·

Basic reporting

English is correct. References are up to date.

Experimental design

The paper is interesting and the results are encouraging, but before publishing, the authors should make a few important corrections. Especially since the methodology and selection of the research has not been explained and is a bit random.
1. The groups are very small, containing 6 animals each. This number is too small for the results to be statistically reliable.
2. The authors tested only one emulsifier. It's a pity that they didn't additionally investigate, for example, carrageenan, which is, it seems, more common in food.
3. The choice of lipocalin-2 test does not seem correct. It is a protein that occurs under physiological conditions
in neutrophil granules, bone marrow, lungs, stomach, small and large intestine, pancreas and kidneys. Lipocalin-2 is mainly studied in kidney diseases. A much better test for the intestine would be calprotectin, which is a typical copromarker. It is a fecal marker useful in detecting inflammation of the digestive tract. Results for calprotectin should be added to the publication.
4. Figures 4 and 5 should be enlarged. The results presented therein are the most important for the research described. Unfortunately, the figures are illegible.
5. It would be worth adding a table summarizing changes in the microbiota of the studied groups.

Validity of the findings

Research is important. However, the studies should be improved, the number of animals tested in groups should be increased and calprotectin tests should be added. This may change the final results.

---

## Round 0.2 · Minor Revisions

After reviewing of your revised submission, one major reviewer has minor comments that upon addressing can be resubmitted

Reviewer 2 ·

Basic reporting

The authors responded to the reviewer's comments and were able to justify their reasons.

Experimental design

However, in supplementary Table 2 you should add a title and description, because it does not mean much. I suggest also adding markings of the increase or decrease in the number of bacteria in a given group.

Validity of the findings

As previously

---

## Round 0.3 · accepted · Accept

Congratulations on the acceptance of your submission for publication with PeerJ. Thank you for working with the reviewers on getting the submission into acceptance shape.